# A Novel Hybrid Drug Delivery System for Treatment of Aortic Aneurysms

**DOI:** 10.3390/ijms21155538

**Published:** 2020-08-02

**Authors:** Koichi Yoshimura, Hiroki Aoki, Chie Teruyama, Masumi Iijima, Hiromori Tsutsumi, Shun’ichi Kuroda, Kimikazu Hamano

**Affiliations:** 1Department of Surgery and Clinical Science, Yamaguchi University Graduate School of Medicine, Ube 755-8505, Japan; kimikazu@yamaguchi-u.ac.jp; 2Graduate School of Health and Welfare, Yamaguchi Prefectural University, Yamaguchi 753-8502, Japan; 3Cardiovascular Research Institute, Kurume University, Kurume 830-0011, Japan; haoki@med.kurume-u.ac.jp; 4Graduate School of Medicine, Yamaguchi University, Ube 755-8611, Japan; k046uh@yamaguchi-u.ac.jp; 5Department of Nutritional Science and Food Safety, Faculty of Applied Bioscience, Tokyo University of Agriculture, Tokyo 156-8502, Japan; mi206786@nodai.ac.jp; 6The Institute of Scientific and Industrial Research, Osaka University, Ibaraki 567-0047, Japan; skuroda@sanken.osaka-u.ac.jp; 7Department of Applied Chemistry, Graduate School of Sciences and Technology for Innovation, Yamaguchi University, Ube 755-8611, Japan; tsutsumi@yamaguchi-u.ac.jp

**Keywords:** drug delivery system, aortic aneurysm, endovascular aneurysm repair, bio-nanocapsule

## Abstract

Ongoing aortic wall degeneration and subsequent aneurysm exclusion failure are major concerns after an endovascular aneurysm repair with a stent-graft. An ideal solution would be a drug therapy that targets the aortic wall and inhibits wall degeneration. Here, we described a novel drug delivery system, which allowed repetitively charging a graft with therapeutic drugs and releasing them to the aortic wall in vivo. The system was composed of a targeted graft, which was labeled with a small target molecule, and the target-recognizing nanocarrier, which contained suitable drugs. We developed the targeted graft by decorating a biotinylated polyester graft with neutravidin. We created the target-recognizing nanocarrier by conjugating drug-containing liposomes with biotinylated bio-nanocapsules. We successfully demonstrated that the target-recognizing nanocarriers could bind to the targeted graft, both in vitro and in blood vessels of live mice. Moreover, the drug released from our drug delivery system reduced the expression of matrix metalloproteinase-9 in mouse aortas. Thus, this hybrid system represents a first step toward an adjuvant therapy that might improve the long-term outcome of endovascular aneurysm repair.

## 1. Introduction

Abdominal aortic aneurysm (AAA) is a noteworthy disease that causes segmental expansion and aortic rupture [1,2]. AAA is characterized by chronic inflammation and progressive extracellular-matrix destruction, by proteolytic enzymes, like matrix metalloproteinases (MMPs), which eventually lead to fatal rupture [3,4,5]. MMP-9 is the primary enzyme responsible for aortic wall degradation. We previously reported that pharmacologic treatment with hydroxymethylglutaryl-coenzyme A reductase inhibitors (statins) inhibited the secretion of MMP-9, a marker of vessel wall degeneration, from human aneurysm tissues in culture [6]. Other clinical studies reported that statins might be associated with the attenuation of aneurysm growth [7,8,9,10]. We also showed in a mouse model that pharmacologic inhibition of c-Jun N-terminal kinase (JNK), a proinflammatory signaling molecule, could successfully treat aortic aneurysms [11,12]. These data and other reports have indicated the potential role of pharmacologic therapy in the treatment of AAA. 

There are two ways to apply drugs in the treatment of AAA: primary therapy and adjuvant therapy. We have long awaited a primary pharmacologic treatment for small AAAs to prevent progression; currently, there is no effective therapy for patients with small AAA. However, large AAAs can be treated with endovascular aneurysm repair (EVAR), and this approach might benefit from adjuvant pharmacologic therapy [13]. 

EVAR has become widely accepted as a minimally invasive treatment for aortic aneurysms [14]. However, after EVAR, occasionally, a late failure occurs in the aneurysm exclusion, which results in aneurysm expansion and rupture [1,15]. Aneurysm progression is caused by aortic wall degeneration. Therefore, an adjuvant pharmacologic intervention that stabilizes the aortic wall might prevent late EVAR failure [13]. A previous randomized clinical trial was conducted to test doxycycline after EVAR. In that trial, doxycycline therapy showed beneficial effects in several patient groups, which suggested that pharmacologic treatment might be useful as an adjuvant therapy to improve EVAR results [16]. However, systemic administration of doxycycline, or other drugs, could cause systemic adverse effects. Moreover, drugs with poor water solubility have limited clinical applications. Alternatively, drug-eluting stent grafts can be used to deliver drugs in combination with EVAR. With this technology, sufficiently high drug concentrations can be delivered to the aortic wall. However, once a stent graft is placed in the body, it is not possible to adjust the drug elution rate.

To address this problem, we created a new hybrid device that combined a drug delivery system and an endovascular stent graft, for treating aortic aneurysms. We called it a rechargeable drug delivery system (RDDS), because it can be charged with a drug, release the drug, and then be recharged, which provided great flexibility in drug administration. Briefly, the RDDS is composed of a target-recognizing nanocarrier that transports the drug to a targeted device (Figure 1A). Any artificial device, like a prosthetic vascular graft, can serve as a targeted device by adding target molecules to the surface. Then, after the device is implanted in the body, the target-recognizing nanocarriers can bind to the target molecules on the device. Utilizing the interaction between biotin and neutravidin, we developed a target-recognizing nanocarrier labeled with biotin and a targeted vascular graft labeled with neutravidin. Most drug types can be incorporated into the nanocarrier. Once the target-recognizing nanocarriers are filled with a suitable drug, they are administered intravenously. The nanocarriers circulate in the bloodstream, then bind to the target molecules on the vascular graft. Next, the nanocarrier shell undergoes hydrolysis, which results in the local delivery of the drug. After the nanocarrier is degraded, the target molecules are regenerated, and thus, they are available for binding to another set of nanocarriers (Figure 1B). 

In the present study, we aimed to demonstrate the feasibility of the RDDS in vitro and in vivo.

## 2. Results and Discussion

### 2.1. Development of the Targeted Graft

First, we intended to label a prosthetic vascular graft with biotin. Initially, we biotinylated a woven polyester graft with an amine-coupling reaction (Figure 2A). Later, we developed a different biotinylated graft by coating a woven polyester graft with a biocompatible polymer, poly (2-hydroxyethyl methacrylate), which could be biotinylated, p(HEMA-biotin) (Figure 2B). The grafts biotinylated with amine-coupling contained 3.1 nmol/cm^2^ biotin, and the grafts biotinylated with a pHEMA coating contained 245 nmol/cm^2^ biotin. Thus, the pHEMA-coating technique greatly increased the amount of biotin on the graft (Figure 2C).

### 2.2. Feasibility of Using the Targeted Graft in Mouse Blood Vessels

We investigated the feasibility of using the targeted graft in blood vessels in vivo. First, we placed the targeted graft, which was coated with p (HEMA-biotin), into the mouse inferior vena cava. Then, we injected fluorescence-labeled neutravidin intravenously. At 15 min after injection, neutravidin had successfully accumulated on the biotinylated graft. This result indicated that the interaction between biotin and neutravidin was preserved within blood vessels in vivo (Figure 3).

### 2.3. Development of the Bio-Nanocapsule-Liposome Complex

The bio-nanocapsule-liposome (BNC-LP) complex was previously reported to be useful for delivering drugs to specific tissues in vivo [17,18,19]. Therefore, we developed a BNC-LP complex as a target-recognizing nanocarrier. First, we selected two drugs for treating aortic aneurysms: SP60015, a JNK inhibitor, and pitavastatin, a statin, and we incorporated them into liposomes (Figure 4A,B). Both SP600125 and pitavastatin were encapsulated within liposomes at favorable concentrations (3.0 mg/mg lipid and 0.37 mg/mg lipid, respectively). The BNC, which is a hepatitis B virus surface antigen L protein with lipid bilayer, was biotinylated (Figure 4C), as described previously [20,21,22]. We then conjugated the biotinylated BNC with drug-containing liposomes to create the drug-containing biotinylated BNC-LP complex, as described previously [17].

Next, we checked the binding of the biotinylated BNC-LP complex to the targeted graft, in vitro. The targeted graft was prepared by allowing DyLight488-labeled neutravidin to bind to the biotin attached to the graft (Figure 4D). Then, we demonstrated that the Cy3-labeled biotinylated BNC-LP complex could specifically bind to the neutravidin attached to the targeted graft in vitro (Figure 4D,E). 

### 2.4. Accumulation of the Target-Recognizing BNC-LP Complex on the Targeted Graft in Mouse Blood Vessels

We checked that the biotinylated BNC-LP complex could bind to the targeted graft in vivo by placing the targeted graft into the mouse inferior vena cava, then injecting the biotinylated BNC-LP complex intravenously. A non-biotinylated BNC-LP complex was used as the control (Figure 5A). At 180 min after injection, the targeted grafts had accumulated the biotinylated BNC-LP complex, but not the control BNC-LP complex (Figure 5B). 

### 2.5. Efficiency of Loading (Charging) the Targeted Graft with the Target-Recognizing BNC-LP Complex In Vivo

Next, we examined how efficiently the target-recognizing BNC-LP complex could be loaded onto the targeted graft. Here, we placed either the targeted graft or an untreated graft into the mouse inferior vena cava. The next day, we injected the biotinylated BNC-LP complex labeled with Cy3 intravenously. At 60 min after injection, the grafts were excised, and we examined the Cy3 fluorescence intensities (Figure 6A). 

We found that the targeted grafts showed significantly higher fluorescence intensities (relative intensity) than the untreated grafts (targeted graft, 1.00 ± 0.11; untreated graft, 0.09 ± 0.01, *p < 0.01* compared to untreated graft). This result indicated that charging the targeted graft was over ten times more efficient than charging an untreated graft (Figure 6B, C).

### 2.6. Efficiency of Recharging the Targeted Graft with the Target-Recognizing BNC-LP Complex In Vivo

Next, we examined the capacity of the targeted graft to be recharged with the target-recognizing BNC-LP complex. We placed the targeted graft into the mouse inferior vena cava. For this recharging experiment, we first intravenously injected a biotinylated BNC-LP complex without the Cy3 label, on the same day of graft placement (Figure 7A). Then, 24 h later, we injected another dose of the biotinylated BNC-LP complex, but this dose was labeled with Cy3. As a control experiment, at 24 h after graft placement, a single intravenous injection of the biotinylated BNC-LP complex labeled with Cy3 was performed. The grafts were excised at about 24 h after graft placement, and the Cy3 fluorescence intensities of the grafts were examined.

The recharged grafts showed high fluorescence intensities (Figure 7B), comparable to those of the singly charged grafts (initial charge intensity: 1.00 ± 0.11; second charge intensity: 0.95 ± 0.24, *n* = 3; Figure 7C). This result suggested that recharging the targeted graft with the target-recognizing BNC-LP complex was highly efficient in vivo.

### 2.7. Effect of Releasing Drug from the Graft Charged with BNC-LP Complexes

Finally, we examined the effects of releasing drug from the graft charged with drug-containing BNC-LP complexes. For this experiment, we prepared a drug-containing graft by combining the targeted graft with the pitavastatin-containing BNC-LP complex. We stimulated the abdominal aorta with 0.5M CaCl_2_ and then placed the pitavastatin-containing graft close to the aorta (Figure 8A). As a control, we used the targeted graft charged with BNC-LP complexes that did not contain a drug (Figure 8B).

In the control experiment, at 24 h after treatment with 0.5M CaCl_2_, MMP-9 was highly expressed in aortic tissues (Figure 8C). Notably, when the pitavastatin-containing graft was placed next to the CaCl_2_-treated aorta, the MMP-9 expression level was significantly reduced (63% ± 9% reduction, *p < 0.05* compared to control, Figure 8D). This result demonstrated that experimental inflammation of mouse aortic tissues was successfully inhibited by the drug released from our hybrid drug delivery system.

### 2.8. Summary of the Results

In summary, the targeted graft was successfully prepared by combining the biotinylated graft with neutravidin. The target-recognizing nanocarrier was created by conjugating biotinylated BNCs with liposomes that contained drugs, such as SP60015 and pitavastatin. Both in vitro and in vivo, the biotinylated BNC-LP complex successfully bound to the targeted graft, but not to an untreated graft. After the target-recognizing BNC-LP complex was intravenously injected, it specifically and effectively accumulated at the targeted graft in the mouse blood vessel. In a recharging experiment, the target-recognizing BNC-LP complex accumulated again at the previously charged targeted graft. These findings indicated that the target-recognizing nanocarriers could charge and recharge the targeted graft in vivo. Finally, the targeted graft charged with pitavastatin-containing BNC-LP complexes significantly reduced MMP-9 expression in aortic tissue, which indicated that the drug had been successfully released from the graft and had treated the aortic wall in vivo. Thus, we successfully developed a novel drug delivery device system, called RDDS. Although the scale of our experimental model system with mice was too small to test the RDDS through every step of the treatment, it was sufficient for testing the system at three stages (charging, releasing, and recharging). Importantly, no mouse died, during this study, due to use of the RDDS, which included biotin, neutravidin, BNC-LP, and pitavastatin.

### 2.9. Clinical Implications and Future Directions

Regardless of promising results in preclinical studies, to date, no AAA drug has shown beneficial effects in the clinical setting. One potential explanation for this failure might be that inappropriate doses were used in previous clinical trials [13,23]. Since AAAs are predominately localized to a limited site on the aorta, it is reasonable to strive for local drug delivery to increase the therapeutic efficacy and reduce systemic side effects. Recently, several studies have demonstrated the efficacy of nanoparticle therapies for treating AAAs in rodent models [24,25,26,27,28]. Those approaches could provide attractive strategies for inhibiting AAA progression. Moreover, other studies have reported the effectiveness of prolonged drug release from biodegradable systems for treating AAAs [29,30,31]. Although drug-eluting stents, in combination with EVAR, might also be a means to deliver drugs to aortic aneurysms, they are likely to lack control of drug elution. In contrast, the RDDS that we developed is distinctively different from systemic drug delivery or drug-eluting stent approaches, and theoretically, the RDDS has more potential for providing great flexibility in drug administration. The RDDS could enable the safe delivery of the desired drugs to an aneurysm, based on a therapeutic marker, when necessary (Figure 9). This approach could counteract aneurysm exclusion failures and encourage AAA regression. Although further studies are necessary before the RDDS can be put into practical use, the system could provide a useful adjuvant therapy to improve the long-term results of EVAR.

## 3. Materials and Methods 

### 3.1. Biotinylation of a Graft Surface and In Vitro Detection of Biotin

The surface of an artificial blood vessel (UBE woven polyester graft, Ube Industries, Tokyo, Japan) was treated with 1 N sodium hydroxide, which partially hydrolyzed it and exposed free carboxyl groups. The carboxyl groups were activated with 1-ethyl-3-(3-dimethylaminopropyl) carbodiimide hydrochloride (EDC), a cross-linker, combined with N-hydroxysuccinimide (NHS). Then, the EZ-Link Biotin-PEO3-LC Amine kit (Pierce, Rockford, IL, USA) was used to attach biotin to the graft surface. 

To produce the p (HEMA-Biotin) conjugate, we performed a condensation reaction between poly (2-hydroxyethyl methacrylate) (pHEMA) and (+)-biotin. We combined pHEMA and biotin at a molar ratio of 1:5, with EDC in N,N-dimethylformamide (DMF), and incubated the reaction at room temperature for 24 h. Then, a graft was immersed in the p (HEMA-Biotin) solution for 6 h to produce a surface-biotinylated graft. The amount of biotin coupled to the graft surface was measured with the 4’-hydroxyazobenzene-2-carboxylic acid biotin quantification kit. For binding assays, the biotinylated graft was combined with DyLight488-labeled neutravidin (Pierce), which served as the target molecule that could bind a biotinylated drug carrier.

### 3.2. Verification of the In Vivo Interaction Between Biotin and Neutravidin

Male C57BL/6 mice, obtained from Chiyoda Kaihatsu (Japan), were used for experiments at 10- to 15-weeks old. An operating microscope with 25× magnification was used for the procedure.

A mid-line abdominal incision was made. The inferior vena cava, together with the infrarenal aorta, were dissected as a unit and mobilized at levels between the renal arteries and the aortic bifurcation. The proximal and distal portions of the inferior vena cava, together with the infrarenal aorta, were clamped as a unit with an Acland microvascular clamp (B-1V). A longitudinal incision was made in the inferior vena cava between the clamps, and a tiny piece of the biotinylated graft (4 × 1 mm) was placed within the lumen of the inferior vena cava. The incision in the inferior vena cava was closed with the interrupted suture technique, performed with 10-0 nylon sutures (10V43-10R, Keisei Medical, Tokyo, Japan). The clamps were then released, and blood flow was reestablished.

Shortly thereafter, DyLight549-labeled neutravidin (Pierce) was injected through the iliac vein. The fluorescent signal was visualized with a fluorescence stereomicroscope (MVX10, Olympus, Tokyo, Japan) in real time. During the fluorescence-labeled neutravidin injection, successful reestablishment of the blood flow was confirmed when the fluorescent signal passed through the inferior vena cava. The interaction between biotin and neutravidin was considered successful when the fluorescent signal accumulated in the biotinylated graft.

All experiments in this study were performed in accordance with the Guidelines for the Care and Use of Laboratory Animals, published by the United States National Institutes of Health. All protocols were approved by the Institutional Animal Care and Use Committee of Yamaguchi University Graduate School of Medicine (No. 31-072, 01/09/2009).

### 3.3. Drug Encapsulation in Liposomes

We used two commonly known therapeutic drugs for aortic aneurysms [6,11,12,32]: a JNK inhibitor (SP600125, Tocris Bioscience, Bristol, UK) and statin (pitavastatin, Santa Cruz Biotechnology, Dallas, TX, USA). We encapsulated these drugs in liposomes. Briefly, lipids (10 mg of dipalmitoylphosphatidylcholine:dipalmitoylphosphatidylethanolamine:dipalmitoylphosphatidylglycerol:cholesterol, at a ratio of: 15:15:40:30) were dissolved in a methanol/chloroform solution (2 mL). Then, the drug mixture (0.5 mg) was dissolved in this solution. The solvent was evaporated in an evaporator and heated in a water bath at 60 °C, to prepare a lipid film. The film was hydrated with a buffer (10 mM 4-(2-hydroxyethyl)-1-piperazineethanesulfonic acid (HEPES), 150 mM NaCl, pH 7.4, 1 mL). Next, the hydrated film was passed through an extruder (500 mL syringe-type, pore size 100 nm, Avestin, Ottawa, ON, Canada) 50 times. Then, gel filtration (Superdex G-50, GE Healthcare, Amersham, UK) was carried out to remove unencapsulated drugs, and thus, purified liposomes were obtained. Both SP600125 and pitavastatin exhibit specific fluorescence characteristics. Therefore, to test fluorescence measurements for detecting the released drugs, purified liposomes were destroyed in the presence of 0.1 N HCl and 0.5% sodium dodecyl sulfate (SDS). 

### 3.4. Preparation of the Biotinylated BNC-LP Complex

As described previously [20,33,34], BNCs were overexpressed in *Saccharomyces cerevisiae* AH22R^−^ cells that carried the ZZ-BNC expression plasmid, pGLD-ZZ50. Next, as described previously [20,35], BNCs were extracted by disrupting the cells with glass beads; then, BNCs were purified on an AKTA chromatography system (GE Healthcare). Next, BNCs were biotinylated with the EZ-Link Sulfo-N-hydroxysuccinimide-biotin kit (Pierce), according to the manufacturer’s protocol. For binding assays, biotinylated BNCs were labeled with Cy3-dye (GE Healthcare) with N-hydroxysuccinimide chemistry, as described previously [36]. Finally, the drug-containing, biotinylated BNC-LP complex was prepared by conjugating the biotinylated BNC with drug-containing liposomes at a weight ratio of 1:35, as described previously [17].

### 3.5. Binding Assay In Vitro

The targeted graft was prepared by incubating the biotinylated graft with DyLight488-labeled neutravidin (0.1 mg/mL)/phosphate buffered saline (PBS) for 30 min. Next, pieces of the targeted graft (5 × 5 mm) were immersed in Tris-buffered saline with Tween20 (TBS-T), and then incubated with 5% bovine serum albumin/TBS-T for 30 min, to block non-specific binding. After washing in TBS-T, the targeted grafts were incubated with the biotinylated BNC-LP complex labeled with Cy3 (0.2 mg/mL)/PBS for 20 min. The targeted grafts were washed three times with TBS-T, then fluorescence was measured with a fluorescence stereomicroscope. Negative controls included non-biotinylated grafts, biotinylated grafts without neutravidin, and non-biotinylated BNC-LP complexes.

### 3.6. Binding Assay In Vivo

The targeted graft was prepared by incubating the biotinylated graft with neutravidin (0.1 mg/mL)/PBS for 30 min. After non-specific blocking and washing, as described in Section 3.5, a piece of the targeted graft (4 × 1 mm) was placed within the lumen of the mouse inferior vena cava. After reestablishing blood flow, we injected the biotinylated BNC-LP complex, labeled with Cy3 (0.2 mg/mL)/PBS (250 μL) into the iliac vein. We used non-biotinylated BNC-LP complex labeled with Cy3 (0.2 mg/mL)/PBS (250 μL) in a control experiment. At 180 min after injection, the mice were sacrificed. The graft was immediately excised, together with the inferior vena cava and infrarenal aorta, as a unit. The specimen was washed with normal saline and examined with a fluorescence stereomicroscope.

### 3.7. Assay for Charging the Targeted Graft with the Target-Recognizing BNC-LP Complex In Vivo

The targeted graft was prepared by incubating the biotinylated graft with neutravidin (0.1 mg/mL)/PBS for 90 min. After non-specific blocking and washing (Section 3.5), a piece of the targeted graft (4 × 1 mm) was placed within the lumen of the mouse inferior vena cava. Non-biotinylated grafts were used as negative controls. At 24 h after placing the graft, the biotinylated BNC-LP complex, labeled with Cy3 (0.025 mg/mL)/PBS (200 μL), was injected into the iliac vein. At 60 min after injection, the mice were sacrificed. The graft was immediately excised, together with inferior vena cava and infrarenal aorta, as a unit. The specimen was washed with normal saline and examined with a fluorescence stereomicroscope.

### 3.8. Assay for Recharging the Targeted Graft with the Target-Recognizing BNC-LP Complex In Vivo

The targeted graft was prepared by incubating the biotinylated graft with neutravidin (0.1 mg/mL)/PBS for 90 min. After non-specific blocking and washing (Section 3.5), a piece of the targeted graft (4 × 1 mm) was placed within the lumen of the mouse inferior vena cava. For this experiment, on the same day, after reestablishing blood flow, the biotinylated BNC-LP complex, without Cy3 label, was injected into the iliac vein (first charge). Then, 24 h after the graft placement, the biotinylated BNC-LP complex labeled with Cy3 (0.025 mg/mL)/PBS (200 μL) was also injected into the iliac vein (recharge). For the control experiment, at 24 h after graft placement, a single intravenous injection of the biotinylated BNC-LP complex labeled with Cy3 (0.025 mg/mL)/PBS (200 μL). At 60 min after injecting the biotinylated BNC-LP complex labeled with Cy3, the mice were sacrificed. The graft was immediately excised, together with the inferior vena cava and infrarenal aorta, as a unit. The specimen was washed with normal saline and examined with a fluorescence stereomicroscope.

### 3.9. Assay for Releasing Drug from the Graft Charged with BNC-LP Complex In Vivo

The targeted graft was prepared by incubating the biotinylated graft with neutravidin (0.1 mg/mL)/PBS for 90 min. The pitavastatin-containing biotinylated BNC-LP complex was prepared by conjugating biotinylated BNC with pitavastatin-containing liposomes (drug content: 0.37 mg/mg lipid, average particle diameter: 577 nm, polydispersity index: 0.399) at a weight ratio of 1:35, as described previously [17]. Subsequently, the pitavastatin-releasing graft was charged by incubating the targeted graft (5 × 5 mm) with the pitavastatin-containing biotinylated BNC-LP complex (0.4 mg)/buffer (0.24 mL of 10 mM HEPES, 150 mM NaCl, pH 7.4) for 90 min, then washing in TBS-T three times. For the control experiment, a control graft was charged by incubating the targeted graft (5 × 5 mm) with the biotinylated BNC-LP complex without drug for 90 min, and then washing in TBS-T three times.

Eight-week-old male C57BL/6 mice were used for this experiment. A mid-line abdominal incision was made. The infrarenal aorta was dissected and mobilized at levels between the renal arteries and the aortic bifurcation. Next, the mobilized aorta was stimulated by applying periaortic 0.5 M CaCl_2_ for 15 min, to induce inflammation in the aorta, as described previously [11,37,38,39]. Then, a piece of the pitavastatin-releasing graft (5 × 1.2 mm) was placed near the part of the abdominal aorta that was stimulated with CaCl_2_. The control graft was used in the control group. The mice were sacrificed at 24 h after the CaCl_2_ treatment. The abdominal aorta was immediately excised and subjected to protein analysis.

Protein extraction and western blotting were performed, as described previously [11,40,41]. Briefly, equal amounts of sample proteins were loaded onto individual lanes in sodium dodecyl sulfate polyacrylamide gels. Then, the proteins were separated by electrophoresis and transferred onto polyvinylidene difluoride membranes. Membranes were probed with antibodies against MMP-9 (1:1,000; R&D Systems, Minneapolis, MN, USA) and glyceraldehyde 3-phosphate dehydrogenase (GAPDH) (1:50,000; Millipore, Billerica, MA, USA).

### 3.10. Statistical Analysis

Data are expressed as the mean ± standard deviation. Statistical analyses were performed with the Student’s *t*-test (*n* = 3) or the Mann–Whitney test (*n* > 3). Data were analyzed with Prism 6.0 (GraphPad Software, La Jolla, CA, USA). *p*-values < 0.05 were considered statistically significant.

## 4. Conclusions

In conclusion, we provided a proof-of-concept for the novel RDDS. We showed that the RDDS was capable of transporting and releasing desired drugs to a specific site, repeatedly and safely. The practical development of this system might be a major step toward improving the long-term outcome after EVAR.

## Figures and Tables

**Figure 1 ijms-21-05538-f001:**
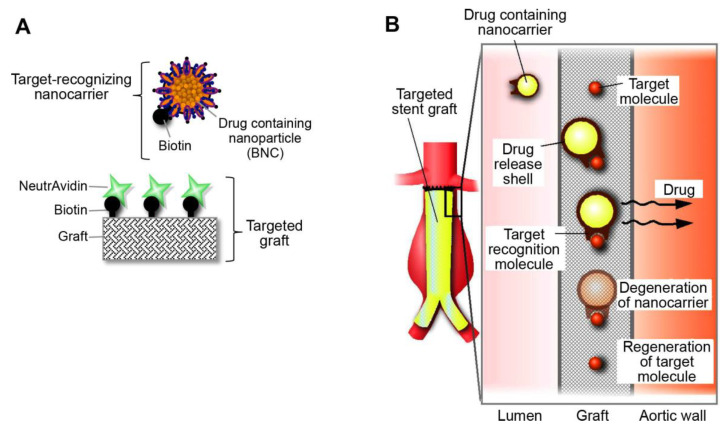
Concept of a novel hybrid drug delivery system. (**A**) Diagram shows the configuration of the system. The targeted graft is decorated with biotin–neutravidin complexes. Neutravidin serves as the target. (**B**) Diagram shows the function of the system. The system is charged by injecting drug-containing nanocarriers. In the aorta, nanocarriers recognize and bind to the target molecules attached to the graft. After releasing the drug at the site of the aneurysm, the nanocarrier degrades, and the target molecules are regenerated.

**Figure 2 ijms-21-05538-f002:**
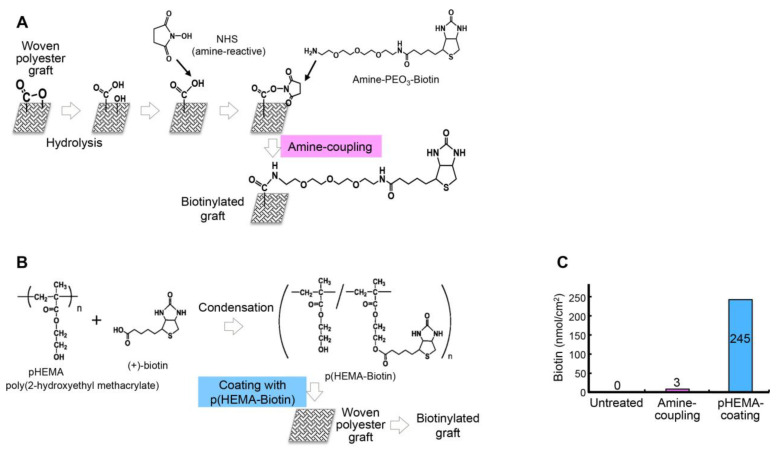
Development of the biotinylated graft. (**A**) Schematic diagram illustrates the preparation of the biotinylated graft with an amine-coupling reaction. (**B**) Schematic diagram illustrates the preparation of the biotinylated graft by coating with poly (2-hydroxyethyl methacrylate) (pHEMA). (**C**) Quantification of the amounts of biotin on the graft surfaces.

**Figure 3 ijms-21-05538-f003:**
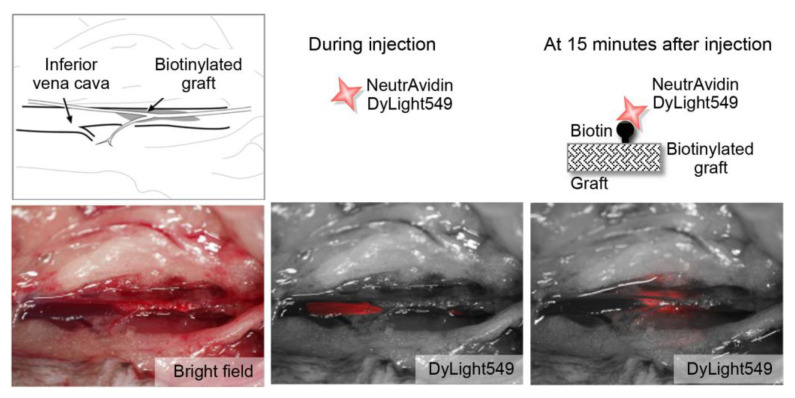
Feasibility of using the targeted graft in mouse blood vessels. (Upper panels) Schematic diagrams show the main features of the procedure. (Lower, left) Bright field image shows the placement of the biotinylated graft in the mouse inferior vena cava. (Lower, middle) Fluorescence image acquired just after injection shows the passage of DyLight549-labeled neutravidin (red) through the inferior vena cava. (Lower, right) Fluorescence image acquired at 15 min after injection shows the accumulation of DyLight549-labeled neutravidin at the biotinylated graft.

**Figure 4 ijms-21-05538-f004:**
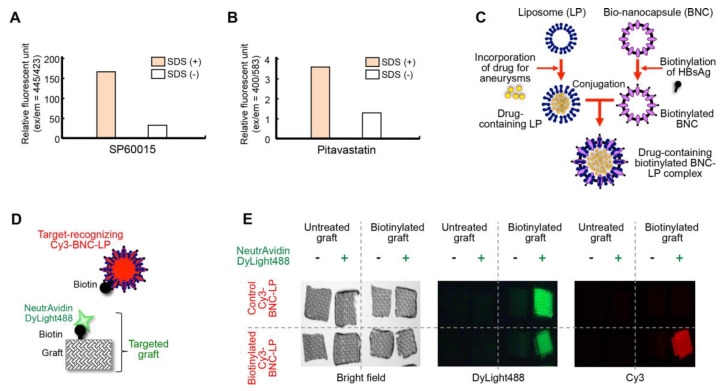
Development of the bio-nanocapsule-liposome (BNC-LP) complex. (**A, B**) Incorporation of (**A**) SP60015 and (**B**) pitavastatin into liposomes. Successful incorporation was determined by measuring drug autofluorescence, before and after liposomes were dissolved with sodium dodecyl sulfate (SDS) treatment. (**C**) Schematic diagram illustrates the preparation of the drug-containing biotinylated BNC-LP complex. The liposome is an empty lipid micelle that can be filled with the desired drug. The bio-nanocapsule is a hollow sphere composed of hepatitis B virus surface antigen (HBsAg). Biotin reacted with HBsAg, then the BNC was conjugated to the LP. (**D**) Diagram shows the configuration of a BNC-LP complex labeled with Cy3. The attached biotin recognizes the neutravidin on the targeted graft. (**E**, left) Bright field images show the woven polyester grafts used for the in vitro binding assay. (Middle) Fluorescence images show the binding of DyLight488-labeled neutravidin (green) to the biotinylated grafts. (Right) Fluorescence images show the binding of the biotinylated Cy3-labeled BNC-LP (red) to the neutravidin on the biotinylated grafts.

**Figure 5 ijms-21-05538-f005:**
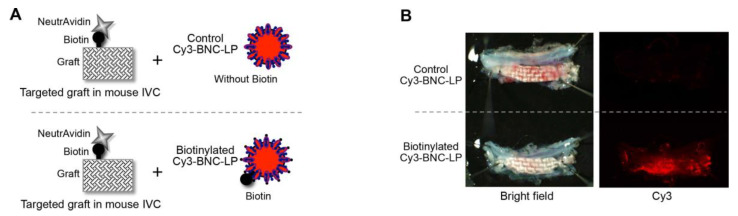
Accumulation of the target-recognizing BNC-LP complex on the targeted graft in mouse blood vessels. (**A**) Schematic diagram of the experiment: the biotinylated BNC-LP complex is compared to the non-biotinylated control. (**B**, left) Bright field images show the grafts excised from mice after the in vivo binding assay. (Right) Fluorescence images show the accumulation of biotinylated Cy3-labeled BNC-LP (red) on the graft.

**Figure 6 ijms-21-05538-f006:**
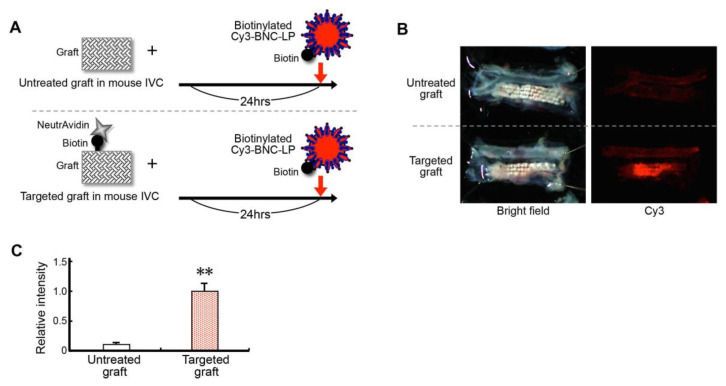
Efficient charging of the targeted graft with the target-recognizing BNC-LP complex in vivo. (**A**) Schematic diagram of the experiment: the targeted graft is compared to the untreated control graft. (**B**, left) Bright field images show the grafts excised from mice after the experiment. (Right) Fluorescence images show the accumulation of Cy3-labeled BNC-LP on the targeted graft. (**C**) Efficiency of charging the targeted graft with the BNC-LP complex, determined by measuring the relative fluorescence intensity. Data are the means ± standard deviations of 3 independent observations. ** *p* < 0.01 compared to the untreated graft.

**Figure 7 ijms-21-05538-f007:**
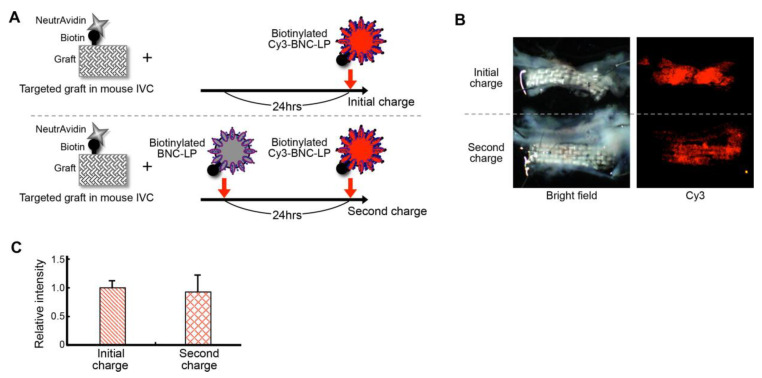
Efficient recharging of the targeted graft with the target-recognizing BNC-LP complex in vivo. (**A**) Schematic diagram of the experiment: an initial charge is compared to a second charge (recharge). (**B,** left) Bright field images show the grafts excised from mice after the experiment. (Right) Fluorescence images show the accumulation of Cy3-labeled BNC-LPs on both targeted grafts. (**C**) Efficiency of recharging the targeted graft with the BNC-LP complex, determined by measuring relative fluorescence intensities. Data are the means ± standard deviations of 3 independent observations.

**Figure 8 ijms-21-05538-f008:**
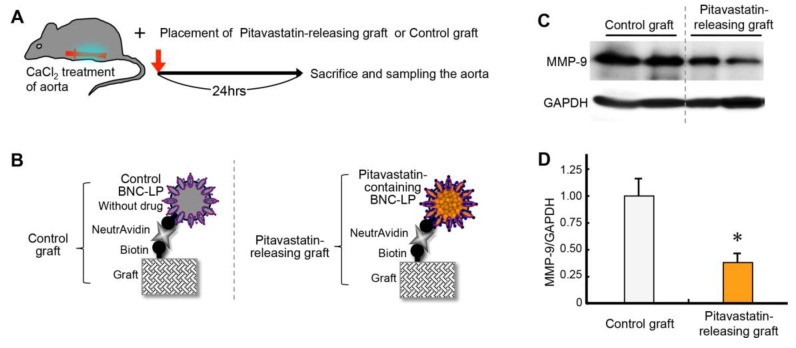
Effect of releasing drug from the graft charged with BNC-LP complexes on mouse aortic tissues. (**A**) Schematic diagram of the experimental design: CaCl_2_ induces inflammation in the aorta (red); the pitavastatin-releasing graft or the control graft is placed next to the aorta. (**B**) Schematic diagram of the experiment: the pitavastatin-releasing graft is compared to the control graft. (**C**) Representative images of western blots for estimating the expression of matrix metalloproteinase-9 (MMP-9) relative to glyceraldehyde 3-phosphate dehydrogenase (GAPDH) expression (internal loading control). (**D**) Quantification of MMP-9 expression in mouse aortic tissues. Data are the means ± standard deviations of 5 independent observations. * *p* < 0.05 compared to the control.

**Figure 9 ijms-21-05538-f009:**
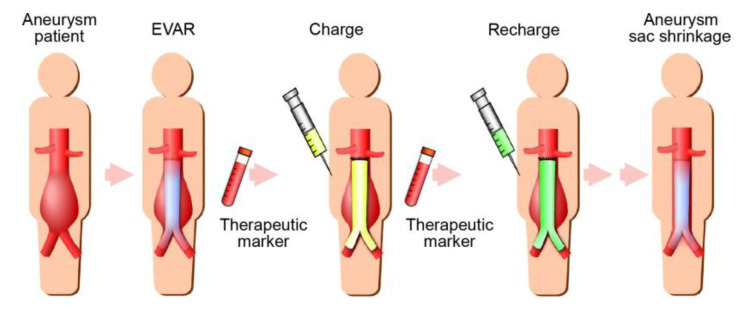
Schematic diagram of future directions for the novel hybrid drug delivery system. The system would enable the delivery of suitable drugs at aneurysms, based on a therapeutic marker when necessary. This system could serve as an adjuvant therapy to improve the long-term outcome of endovascular aneurysm repair (EVAR).

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
