# Peer review of "A Novel Hybrid Drug Delivery System for Treatment of Aortic Aneurysms"

_ijms, 2020, doi:10.3390/ijms21155538_

Round 1

Reviewer 1 Report

The model proposes a drug delivery system that rely on the placement of a previously treated stent graft and its interaction with the aortic wall to deliver nano-carriers with drug-containing bio-nano capsules. The system was tested in-vitro, and in-vivo (in mice), showing a reduction of MMP-9 in mice vessel wall.  The study is innovated and highly relevant, most importantly, it address key aspects that can potentially help preventing AAA rupture.

There are some aspects, however, that might limit the impact of this study: 1) Relying on a coating stent graft as mean to deliver drug-nanoparticles. This would limit the treatment solely to patients with large AAAs (according to clinical guidelines for elective surgery), which would help slowing down AAA enlargement but not decreasing the risk rupture  associated to a highly compromise aortic wall that are usually found on these patients.

2) By not considering the fact that at least 75% of AAA detected (Harter et. al. 2005) have a significant layer of intraluminal thrombus (ILT) covering the aortic wall. It has been shown that the ILT content increases at the same rate at which aneurysm expands (Zambrano et. al. 2016). This would narrow even more the use of the proposed therapy.

Related to the experimental to the study experimental design, mayor concerns includes: 1) the low number of subjects used for each analysis, 2) lack of statistical tests (t-tests to suggests significant differences), and 2) the fact that the whole delivery system was not fully tested (i.e. particles being released to the blood, nano-particles biding graft receptors, drug delivering to the wall), nor that an aneurysm animal model was used to assess the effectiveness of the proposed method.

Specific major and minor comments below:

Major:

Line 161: Although the differences are clear, the sample number for this analysis seems too low.  Did the authors performed a t-tests analysis? Also, could you please expand on the experimental design followed to select the minimum number of samples needed to  achieve significant differences? Without a statistical test or a large number of sample results could be misleading

Line 86: what is the size of the nanocarrier? Is it small enough to diffuse through the already covered stent graft?. This could pose an important limitation in delivering the drug. Specially, because the whole system has not been tested.

Line 191: sample number seems too low  to suggests a trend.

Line198: I believe it would be ideal to test the effect of the drug reducing MMP-9 by releasing nano-carriers through the blood instead of coating the stent graft with it.

Minor:

Line 55: Although it has been considered as a minimally  invasive; however, it poses some risk associated to this surgery. Furthermore, due to the age of the patients, not all of them are eligible for the procedure.

Line 82: Although it might be already known, It might be important to add the potential biochemical interaction between biotin-neutravidin and endothelium to inform a more general audience

Line 197: Rephrase the sentence. It was not clear whether the CaCl2 was applied to the graft or the aorta.

Line 204: the experiment showed that the drug in-fact inhibited the inflammation. However, we are still uncertain the benefit of the treatment in an aneurysms case scenario. Specially because an aneurysms model was not used in the analysis.

Line 358:  Is there a reason why for this part of the experimentation SP600125 was not tested?.

Author Response

Responses to Comments by Reviewer 1

General comments

“The model proposes a drug delivery system that rely on the placement of a previously treated stent graft and its interaction with the aortic wall to deliver nano-carriers with drug-containing bio-nano capsules. The system was tested in-vitro, and in-vivo (in mice), showing a reduction of MMP-9 in mice vessel wall. The study is innovated and highly relevant, most importantly, it address key aspects that can potentially help preventing AAA rupture.”

Response: We appreciate the positive comment. We also thank the reviewer for the thoughtful suggestions, which helped us improve our manuscript. We have revised our manuscript in response to the comments, as explained below, on a point-by-point basis.

“There are some aspects, however, that might limit the impact of this study: 1) Relying on a coating stent graft as mean to deliver drug-nanoparticles. This would limit the treatment solely to patients with large AAAs (according to clinical guidelines for elective surgery), which would help slowing down AAA enlargement but not decreasing the risk rupture associated to a highly compromise aortic wall that are usually found on these patients.”

Response: As described in the original manuscript, we developed a rechargeable drug delivery system (RDDS), which provides adjuvant therapy after an endovascular aneurysm repair (EVAR). Therefore, inevitably, the adjuvant therapy with RDDS is only applicable to patients that have been treated with EVAR for large abdominal aortic aneurysms (AAAs). In those cases, the risk of AAA rupture could be avoided in the short term with an initial, successful EVAR, and post-EVAR risk could be reduced over the long term with the adjuvant therapy.

“2) By not considering the fact that at least 75% of AAA detected (Harter et. al. 2005) have a significant layer of intraluminal thrombus (ILT) covering the aortic wall. It has been shown that the ILT content increases at the same rate at which aneurysm expands (Zambrano et. al. 2016). This would narrow even more the use of the proposed therapy.”

Response: Theoretically, the adjuvant therapy with the RDDS could also have a beneficial effect on AAA with intraluminal thrombus (ILT), based on the following two reasons. First, it is reported that ILT is the major source of proteases, such as matrix metalloproteinase-9 (MMP-9), and they are associated with AAA wall weakening (Atherosclerosis 2011, 218, 285–286). Therefore, both the aortic wall and ILT could be good targets for this therapy. Second, the RDDS could deliver drugs to the aortic wall in the neck and transitional areas of the aorta, which contain inflammatory regions; these regions represent a good therapeutic target, due to the presence of relatively little ILT in these areas.

“Related to the experimental to the study experimental design, mayor concerns includes: 1) the low number of subjects used for each analysis, 2) lack of statistical tests (t-tests to suggests significant differences), and 2) the fact that the whole delivery system was not fully tested (i.e. particles being released to the blood, nano-particles biding graft receptors, drug delivering to the wall), nor that an aneurysm animal model was used to assess the effectiveness of the proposed method.”

Response: In response to this reviewer’s suggestion, we performed additional statistical tests and we found a statistically significant difference between the targeted grafts and the untreated grafts. We have added this result to Figure 6C, and we describe it in the revised manuscript: Lines 162-163 of the Results section.

Unfortunately, the scale of our experimental model system with mice was too small to test the RDDS through every step of the treatment. However, it was sufficient for testing the system at three stages (charging, releasing, and recharging). We have added this comment to the revised manuscript: Lines 227-230 of the Discussion section.

Major comments

“Line 161: Although the differences are clear, the sample number for this analysis seems too low. Did the authors performed a t-tests analysis? Also, could you please expand on the experimental design followed to select the minimum number of samples needed to achieve significant differences? Without a statistical test or a large number of sample results could be misleading”

Response: As noted above, we performed additional statistical tests and revealed a statistically significant difference (p<0.01) between the targeted grafts and the untreated grafts. We have added this result to Figure 6C, and we described it in Lines 162-163 of the Results section in the revised manuscript.

“Line 86: what is the size of the nanocarrier? Is it small enough to diffuse through the already covered stent graft?. This could pose an important limitation in delivering the drug. Specially, because the whole system has not been tested.”

Response: Regarding the size of the nanocarriers, the average diameter of the liposomes used for preparing pitavastatin-containing nanocarriers was 577 nm, as described in Line 363. However, conceptually, this system for delivering drugs to the aortic wall would not be affected by the size of the drug-containing nanocarriers, because the drug-containing nanocarrier is designed to biodegrade in the graft, which releases the drug; then, the actual drug substance freely diffuses to the aortic wall through the graft (described in Lines 87-88 and in Figure 1B). Indeed, we demonstrated the effect of releasing drug from the graft charged with drug-containing nanocarriers into an inflamed aortic wall in mice, although the whole system was not tested all the way through.

“Line 191: sample number seems too low to suggests a trend.”

Response: We agree with the reviewer that our description in Lines 192-193 was an overstatement. However, even if we increased the number of subjects, it would not be possible to conclude that the efficiencies of both the initial and second charges are equivalent. Therefore, we changed the phrase: “This result indicated that…” to “This result suggested that…” (Line 192).

“Line198: I believe it would be ideal to test the effect of the drug reducing MMP-9 by releasing nano-carriers through the blood instead of coating the stent graft with it.”

Response: In this study, we successfully demonstrated a proof-of-concept for the RDDS, at a number of steps. As part of this process, the experiment shown in Figure 8 revealed the effect of releasing the drug from the graft charged with drug-containing nanocarriers into the aortic wall. In the experiment suggested by the reviewer, there may be a concern that the effects of drug-containing nanocarriers directly accumulated into aortic wall from bloodstream could not be distinguished from those of drug-containing nanocarriers that accumulated in the graft from bloodstream and then released drug from the graft into the aortic wall.

Minor comments

“Line 55: Although it has been considered as a minimally invasive; however, it poses some risk associated to this surgery. Furthermore, due to the age of the patients, not all of them are eligible for the procedure.”

Response: We agree with the reviewer that EVAR poses some risk, and it is not applicable to all patients with AAA. As noted above, we conducted this study to provide an adjuvant therapy that could be administered after EVAR. Other studies must be conducted to develop a method for treating AAA in patients that are unsuitable for EVAR.

“Line 82: Although it might be already known, It might be important to add the potential biochemical interaction between biotin-neutravidin and endothelium to inform a more general audience”

Response: We may be misinterpreting your point, but we do not think this would be helpful for the general audience, because the RDDS does not theoretically rely on an interaction with the endothelium. Therefore, it does not seem important to explain any potential interaction between biotin-neutravidin and the endothelium.

“Line 197: Rephrase the sentence. It was not clear whether the CaCl2 was applied to the graft or the aorta.”

Response: In response to the reviewer’s comment, we changed the sentence from “We placed the pitavastatin-containing graft close to the abdominal aorta, which had been stimulated with 0.5M CaCl2” to “We stimulated the abdominal aorta with 0.5M CaCl2 and then placed the pitavastatin-containing graft close to the aorta…” in Lines 197-199.

“Line 204: the experiment showed that the drug in-fact inhibited the inflammation. However, we are still uncertain the benefit of the treatment in an aneurysms case scenario. Specially because an aneurysms model was not used in the analysis.”

Response: We agree with the reviewer that we could not conclude that pharmacotherapy with the RDDS was effective in treating AAA in mice, because the RDDS was not yet tested in a mouse model of AAA. However, more importantly, the aim of this study was to establish a proof-of-concept for the RDDS. As part of this aim, we successfully demonstrated that a drug released from the graft charged with drug-containing nanocarriers had a direct effect on inflammation in the aortic wall. This result was sufficient to confirm that the drug-releasing feature of the RDDS was feasible.

“Line 358: Is there a reason why for this part of the experimentation SP600125 was not tested?.”

Response: Yes. Pitavastatin is currently in clinical use; in contrast, the use of SP600125 is limited to experimental studies.

Reviewer 2 Report

 It is a really interesting research article about a novel drug delivery system for minimally invasive  treatment of aortic aneurysms. As authors say further studies are, of course, needed before it can be put into practical use but the system could provide a useful adjuvant therapy to improve   the long-term results of EVAR. The initiative of this type of treatment is very appropriate and appreciated 

Author Response

Responses to Comments by Reviewer 2

General comments

“It is a really interesting research article about a novel drug delivery system for minimally invasive treatment of aortic aneurysms. As authors say further studies are, of course, needed before it can be put into practical use but the system could provide a useful adjuvant therapy to improve the long-term results of EVAR. The initiative of this type of treatment is very appropriate and appreciated”

Response: We thank the reviewer for these kind words and for taking the time to make thoughtful comments.

Round 2

Reviewer 1 Report

First of all, I would like to reiterate that the study is interesting and the proposed technique has a great potential to reduce post-EVAR associated fatalities associated with the AAA wall failure. However, I believe that this manuscript should be considered for publication after the authors address two minor concerns:

-The authors agree that this study is limited to patients that have been treated with EVAR since it would rely on the placement of a stent graft. This procedure itself is not “minimally invasive” as it is suggested in the title of the study.  I suggest to modify the title to more accurately depict the study.

Also, claims that this technique could have beneficial effects on AAAs with ILT. If this is the case, they should discuss the potential of this technique to manage the ILT inside post-EVAR AAAs.

Author Response

Responses to Comments by Reviewer 1

“First of all, I would like to reiterate that the study is interesting and the proposed technique has a great potential to reduce post-EVAR associated fatalities associated with the AAA wall failure. However, I believe that this manuscript should be considered for publication after the authors address two minor concerns:”

Response: We appreciate the constructive comments. We have revised our manuscript in response to the comments, as explained below, on a point-by-point basis.

“The authors agree that this study is limited to patients that have been treated with EVAR since it would rely on the placement of a stent graft. This procedure itself is not “minimally invasive” as it is suggested in the title of the study. I suggest to modify the title to more accurately depict the study.”

Response: In response to this reviewer’s suggestion, we changed the title: “A novel drug delivery system for minimally invasive treatment…” to “A novel hybrid drug delivery system for treatment…” (Lines 2-3).

“Also, claims that this technique could have beneficial effects on AAAs with ILT. If this is the case, they should discuss the potential of this technique to manage the ILT inside post-EVAR AAAs.”

Response: As we mentioned in the previous response letter, the adjuvant therapy with this system could theoretically have a beneficial effect on AAA with ILT. Unfortunately, however, we do not have any experimental data to support the theory. We are sorry, but we would like to include this issue not in this manuscript, but in future studies.